# OpenReview forum: "How Does RLHF Shift Behavior Distributions? Distinguishability and Steerability"
_ICLR.cc/2024/Conference — Submitted to ICLR 2024_

### Official Review · Reviewer_jcwj · 2023-11-01

**Soundness:** 1 poor
**Presentation:** 2 fair
**Contribution:** 1 poor
**Rating:** 3
**Confidence:** 4

**Summary:**

This paper studies how language models (LMs) answer yes/no questions about their likelihood to say a statement before and after RLHF. An analysis is presented on statements grouped by desirable and undesirable behaviors from the Anthropic persona evaluation dataset with respect to the distinguishability of behavior distributions. Finally, prompts of varying lengths are created and prepended to corresponding statements. Experimental results show steerability for these questions increases with prompt length and the RLHF model is more steerable for both positive and negative behaviors.

**Strengths:**

1. Better and more broadly characterizing the changes LMs undergo as a result of fine-tuning procedures like RLHF is an important topic.

**Weaknesses:**

**Weaknesses**

1. The questions asked and analysis performed in this work is very similar to [1, 2]. The paper should clarify contributions and better differentiate what findings are unique.
2. The paper should better motivate the "Is the following statement something you would say?" questions. Section 4.1 states "which reflects the probability of the model will tend to say yes to such a negative statement (and hence behave undesirably)". Section 4.2 states "where a larger value indicates the model’s higher likelihood to assert the input statement". However, no evidence is provided for these statements. Since the interestingness/meaningfulness of the analysis presented is predicated on this assumption, it should be supported with actual evidence. To the best of my knowledge, the ability to self-reflect in this manner has only been demonstrated in certain limited contexts and settings; such as models reflecting on the validity of their own answers post-hoc [3]. However, that setting is quite different from the one here as the factuality of a statement is independent of the model.
3. The methodology in Section 4.2 seems flawed. Unnormalized logits aren't comparable across timesteps/examples since they have different normalization constants.
4. The analysis is limited to a subset of behaviors. This choice is justified because "we select a subset of behaviors whose embodiment by an AI is likely to have pretty broad agreement on whether humans would like or dislike." I'm not particularly convinced by this. I, for one, would be significantly more concerned about current models displaying "anti-LGBTQ-rights" behaviors (which are somewhat trivialized in the paper as being political) than behaviors like "acts-like-it-wants-to-help-humans-but-does-not-care-about-that", as the near-term causal mechanism for potential harm is much clearer to me regarding the former given current model capabilities. Of course, I don't expect to agree with the premises of all research, but I do expect papers to motivate their research program using more rigor than assertions of consensus.

**References**

1. Ethan Perez et al "Discovering Language Model Behaviors with Model-Written Evaluations." 2022
2. Yotam Wolf et al. "Fundamental Limitations of Alignment in Large Language Models." 2023
3. Saurav Kadavath et al. "Language Models (Mostly) Know What They Know." 2022

**Questions:**

N/A

---

> ### Author Response · Authors · 2023-11-17
>
> We sincerely appreciate your feedback and thoughtful comments. We address each point one by one below.
>
> > **W1.** The paper should clarify contributions and better differentiate what findings are unique from Wolf et al. (2023) and Perez et al. (2023).
>
> We attempted to elucidate the distinctions between this paper, Wolf et al. (2023), and Perez et al. (2023) in the **Introduction** and **Related Works section**. Wolf et al. (2023) provide theoretical frameworks which we empirically investigate to see whether it truly manifests in Language Models in a much more extensive manner than their previous version had. Now, it seems they go into more empirical investigation, however it remains relegated to toy scenarios where they separately fine-tune Llama 2 13B into separate positive and negative models. In contrast, we look into individual production models, a much more practical scenario. Perez et al. (2023) provide the persona dataset and evaluate it on a baseline model, but do not evaluate it on a steered model nor do they attempt to correlate characteristics of the behavioral distribution with steerability.
>
> > **W2.** Section 4.2 states "where a larger value indicates the model’s higher likelihood to assert the input statement". However, no evidence is provided for these statements. Since the interestingness/meaningfulness of the analysis presented is predicated on this assumption, it should be supported with actual evidence. To the best of my knowledge, the ability to self-reflect in this manner has only been demonstrated in certain limited contexts and settings; such as models reflecting on the validity of their own answers post-hoc. However, that setting is quite different from the one here as the factuality of a statement is independent of the model.
>
> Indeed, what we are looking at is not the model's true tendency towards saying the statement but rather simply it's likelihood to say that it "would" say that statement. The language here should have been more precise.
>
> > **W3.** The analysis is limited to a subset of behaviors. This choice is justified because "we select a subset of behaviors whose embodiment by an AI is likely to have pretty broad agreement on whether humans would like or dislike." I'm not particularly convinced by this. I, for one, would be significantly more concerned about current models displaying "anti-LGBTQ-rights" behaviors (which are somewhat trivialized in the paper as being political) than behaviors like "acts-like-it-wants-to-help-humans-but-does-not-care-about-that", as the near-term causal mechanism for potential harm is much clearer to me regarding the former given current model capabilities. Of course, I don't expect to agree with the premises of all research, but I do expect papers to motivate their research program using more rigor than assertions of consensus.
>
> We hear your perspective here, and strongly sympathize with it. To clarify, we had hoped to step back and remove ourselves from any controversial topics, especially in light of the alignment we concern ourself with being with respect to the *entirety* of humanity's differing values. We tried to select behaviors that it'd be hard to picture any human disagreeing on, no matter our personal beliefs. To do so, we took dedicated time to parse through the available behaviors in Anthropic's persona evaluation dataset.
>
> Thank you for these suggestions. We will duly take them into account.

---

> > ### Comment · Reviewer_jcwj · 2023-11-18
> > **Request for comment**
> >
> > Thank you for your response. Could you please comment on my original W3 regarding unnormalized logits. This was also noted by Reviewer uiXy (W4) and is potentially a significant issue with the analysis.

---

> > > ### Comment · Reviewer_jcwj · 2023-11-22
> > > **Reply to authors**
> > >
> > > I appreciate the author's response. Unfortunately, several of my concerns remain unaddressed and the authors have not replied to my or other reviewer's requests for specific comments. I will leave my score unchanged.

---

> ### Author Response · Authors · 2023-11-22
>
> We apologize for the late response not in time before your final reply due to our busy schedule.
>
> > The methodology in Section 4.2 seems flawed. Unnormalized logits aren't comparable across timesteps/examples since they have different normalization constants.
>
> We appreciate your suggestion. We will revise the paper with log probability to avoid this normalization issue.

---

### Official Review · Reviewer_uiXy · 2023-11-07

**Soundness:** 1 poor
**Presentation:** 3 good
**Contribution:** 1 poor
**Rating:** 3
**Confidence:** 4

**Summary:**

The paper uses a previously proposed model of "behavior decomposition" to investigate the responses of a large language model (LLaMA2-7B) on the Anthropic's persona evaluation dataset. First, the paper attempts to empirically validate a theoretical claim that RLHF makes the distributions of "positive" and "negative" behaviors more distinguishable. The paper uses two metrics for that: KL-divergence and Wasserstein distance. With the first metric, no consistent increase in distinguishability is observed. With the second, an increase is observed more often. The paper then analyzes "steerability" of the LLM towards misaligned behavior before and after RLHF. Steerability in this context means the ability for the model to exhibit negative persona traits after prompting with persona descriptions of varying length. The model after RLHF is shown to be more steerable overall, and it is shown that it requires shorter prompts. The paper postulates that these results suggest that RLHF is not adequate as an algorithm for aligning LLMs.

**Strengths:**

- The paper follows the important direction of evaluating the psychological characteristics of language models.
- The overview of the related work is rather comprehensive.
- The paper tries to convey the message that a single language model after RLHF will necessarily fail to encompass the diversity of human values. I believe this is an important idea.

**Weaknesses:**

I think the results in the paper could provide a starting point for an investigation, but are currently not sufficient for a conference paper.
- The experiments with distinguishability are inconclusive, i.e. it is unclear whether RLHF significantly increases distinguishability of positive and negative behaviors. Depending on whether $\beta$-distinguishability or Wasserstein distance is used, the results come out differently. The paper does not provide an experimentally supported explanation for this phenomenon. Instead, it makes a vague claim: "We postulate that this inconsistency may stem from the process not being explicitly tailored to target specific human-like psychological behaviors for avoidance." A rigorous evaluation for why the inconsistency exists would be crucial to make the claims in the paper interesting.
- Steerability results are also quite limited. The plots in Figures 3 and 9, for most behaviors, show that there is a single prompt (most often 2 to 4 sentences long) which is sufficient to make the RLHF model understand the persona, and the prompts of higher length do not increase the understanding, so misalignment rate just oscillates stochastically. The base model does not show the high misalignment rate likely because it simply cannot follow the instructions as well as the model after RLHF. Arguably, there is not much correlation between the prompt length and misalignment beyond this simple effect.
- I think impersonating a character with provided traits and answering yes-or-no questions from the character's point of view does not constitute dangerous behavior by the model, so it was not removed by RLHF. The main claim of Section 5 is that "RLHF model is more steerable to exhibit negative behaviors often". A good RLHF algorithm makes a model more steerable in any direction, while prohibiting dangerous behavior. Hence, I suspect that RLHF with more strict criteria provided to evaluators, such as "do not allow the model to impersonate negative characters" could avoid this issue. I think the conclusion that "the prevailing practice of training a centralized RLHF model may not be optimally suited to address the intricate and diverse spectrum of human values, calling on future approaches for more comprehensive alignment techniques" is plausible, but it does not follow from the results of the paper. Why discard RLHF if we can simply adjust the annotation process?
- Figures 2 and 6 show logit values of the "Yes" output. If I understand this correctly, these are the pre-softmax values produced by the LLM. These values are not normalized, so the plots are not very meaningful. The authors observe that "the output values generally shift to be smaller for the RLHF model compared to the base model". This is not an interesting observation, since the softmax output does not change if all inputs are shifted by a constant. A more interesting quantity to look at would be $\log \mathbb{P}[\text{Yes}\mid s]$, where $s$ is the persona prompt and the statement. Footnote on page 5 states that the probabilities can be extremely small, so we should not look at them directly, but $\log$ probabilities would not suffer from this issue. I suspect that if we analyzed the distributions of $\log \mathbb{P}$ among the questions, the Wasserstein distance results would come out as more ambiguous.
- The paper fails to connect the results for distinguishability and steerability: "For most plots, the RLHF model (in orange) is capable of being prompted to behave more negatively with a higher misalignment rate than the base model (in gray). However, the degree to which this occurs does not seem to be predicted by the distinguishability". This failure to conform to the theoretical results from Wolf et al is not explained. I suspect that the reason is that Wolf et al claim that when the behaviors are better distinguishable, there _exists_ a short prompt that induces negative behavior. However, the authors of the paper at review do not search for a prompt of a given length extensively, and only try one prompt for each length.
- The text is rather diluted, with many ideas repeated almost word-for-word multiple times, e.g. the claim that RLHF cannot be used to align language models properly.
- The phrasing "we conceptualize LLM outputs as a decomposition of behaviors into positive and negative sub-distributions" in the abstract is misleading, since the authors are not the first ones to propose this decomposition, as they note in the main text.

**Questions:**

How is the Wasserstein distance measured? For two general distributions it is intractable to compute it, hence some heuristic is needed. The paper does not provide any details on the used heuristic.

---

> ### Author Response · Authors · 2023-11-17
>
> We sincerely appreciate your feedback and insightful comments. We address each point one by one below.
>
> > **W1.** Confusion around $\beta$-distinguishability vs. Wasserstein distance.
>
> Due to space limitations, please see our response to W3 for Reviewer dnKS above.
>
> > **W2.** Prompts of higher length do not increase the understanding, so misalignment rate just oscillates stochastically. The base model does not show the high misalignment rate likely because it simply cannot follow the instructions as well as the model after RLHF. Arguably, there is not much correlation between the prompt length and misalignment beyond this simple effect.
>
> The lack of correlation between prompt length and misalignment beyond the effect described is true. However, we must note that this then stands in contrast to the claims made in Wolf et al. (2023), which state that you can continually recondition the model with additional sentences to shift it further and further into negative behavior. This shows that for LLaMa 2, past a certain point, reconditioning with more sentences doesn't have much of an effect. One can expect that with bigger models, the range that reconditioning with more sentences would grow, but perhaps there too is a certain point where the effects proposed in Wolf et al. (2023) also would decay away, and the theoretical conclusion does not necessarily hold.
>
> > **W3.** I think the conclusion that "the prevailing practice of training a centralized RLHF model may not be optimally suited to address the intricate and diverse spectrum of human values, calling on future approaches for more comprehensive alignment techniques" is plausible, but it does not follow from the results of the paper. Why discard RLHF if we can simply adjust the annotation process?
>
> These are fair points, and our wording was not sufficiently precise in how we addressed this. In retrospect, we should not have argued that RLHF fundamentally could not target human-like psychological behaviors for avoidance. Instead, we should have argued that with how it's currently used by the large majority of practictioners, it does not often do so, which presents a potential vulnerability.
>
> However, even if targeted in the training process, Wolf et al. (2023)'s claims that if the negative subdistributions are shifted down and the positive subdistributions are shifted up, thereby making them more distinguishable, this leaves the model extra vulnerable to adversarial attacks. This is due to the fact that according to their framework, in the minimal case a prompt then must exist of shorter sentence count to make it behave poorly. We end up failing to find evidence in support of this claim, but, if true, this could present a fundamental RLHF limitation like worried about above.
>
> > **W4.** The paper fails to connect the results for distinguishability and steerability. The failure to conform to the theoretical results from Wolf et al is not explained. I suspect that the reason is that Wolf et al claim that when the behaviors are better distinguishable, there exists a short prompt that induces negative behavior.
>
> The suspicion you raise is interesting. However, Wolf et al. (2023)'s claim is not that there exists a short prompt to induce negative behavior. Rather, it is that the more distinguishable distributions are, the fewer sentences it should require to be able to recondition the model towards behaving negatively. The crux is on how many sentences are needed to repeatedly recondition the model, rather than whether a short prompt exists or not. Generally, the short prompt existing is what adversarial attacks usually tend themselves to and is a more discussed issue. However, it is not the claim that Wolf et al. were making.
>
> > **W5.** The text is rather diluted, with many ideas repeated almost word-for-word multiple times, e.g. the claim that RLHF cannot be used to align language models properly.
>
> We repeated ideas to attempt to reinforce them and remind the reader of the broader narrative we were working towards. We apologize if this made the paper feel diluted.
>
> > **W6.** The phrasing "we conceptualize LLM outputs as a decomposition of behaviors into positive and negative sub-distributions" in the abstract is misleading, since the authors are not the first ones to propose this decomposition, as they note in the main text.
>
> Indeed, we should modify the wording to be more precise. Our introduction has elaborated more on this: _"This conceptualization involves decomposing model output distributions into two components or sub-distributions (Wolf et al., 2023)"_, with proper crediting.
>
> Thank you for this great feedback. We will duly take them into account.

---

> > ### Comment · Reviewer_uiXy · 2023-11-20
> > **Connection to results from Wolf et al, logits vs log probabilities**
> >
> > > However, Wolf et al. (2023)'s claim is not that there exists a short prompt to induce negative behavior. Rather, it is that the more distinguishable distributions are, the fewer sentences it should require to be able to recondition the model towards behaving negatively.
> >
> > Quoting from the abstract of Wolf et al (emphasis mine):
> > > Importantly, we prove that within the limits of this framework, for any behavior that has a finite probability of being exhibited by the model, *there exist prompts* that can trigger the model into outputting this behavior, with probability that increases with the length of the prompt.
> >
> > This claim is mirrored in the main text of Wolf et al in formulation and discussion of Theorem 1. Indeed, the paper provides an upper bound on the length of the prompt, and higher distinguishability of behaviors leads to shorter prompts. However, it still gives an existence result, and if the experiments with a single set of prompts do not show the expected misalignment rates, it does not necessarily contradict the theoretical results from Wolf et al. Because of this issue, the claim from the response that
> > > This shows that for LLaMa 2, past a certain point, reconditioning with more sentences doesn't have much of an effect.
> >
> > also does not follow from the experiments.
> >
> >
> > The W numbers in the authors' responses to my and Reviewer jcwj's comments are incorrect, skipping over W4 in my response and over W3 in jcwj's response. Regarding my original W4, I agree with Reviewer jcwj that the logit issue still has to be addressed.

---

> ### Author Response · Authors · 2023-11-22
>
> >**R1.** "However, Wolf et al. (2023)'s claim is not that there exists a short prompt to induce negative behavior. Rather, it is that the more distinguishable distributions are, the fewer sentences it should require to be able to recondition the model towards behaving negatively."
> > Quoting from the abstract of Wolf et al. (emphasis mine):
> >"Importantly, we prove that within the limits of this framework, for any behavior that has a finite probability of being exhibited by the model, there exist prompts that can trigger the model into outputting this behavior, with probability that increases with the length of the prompt."
> > This claim is mirrored in the main text of Wolf et al. in formulation and discussion of Theorem 1. Indeed, the paper provides an upper bound on the length of the prompt, and higher distinguishability of behaviors leads to shorter prompts. However, it still gives an existence result, and if the experiments with a single set of prompts do not show the expected misalignment rates, it does not necessarily contradict the theoretical results from Wolf et al. Because of this issue, the claim from the response that
> > "This shows that for LLaMa 2, past a certain point, reconditioning with more sentences doesn't have much of an effect.""
> > Also does not follow from the experiments.
>
> As you've shown, they do provide a claim that the existence of a prompt to misalign has increasing probability as the length of the prompt increases. Rather than directly investigating this claim and attempting to find this prompt, what we attempted to look at was whether the mechanism used in their proof derivation for this claim seemed to manifest. In particular, whether every additional sentence indeed pushed the model towards behaving more negatively.
>
> Unfortunately, as you gestured towards, these shifts were on average (sampling from negative sub-distribution and taking the KL-Divergence) rather than in the minimal case. As such, our use of only one set of negative prompts rather than sampling many prompts from the negative distribution was suboptimal. We could not perform the latter because we were working with individual production models rather than separate toy positive and negative models as Wolf et al. did. In other words, we had no negative distribution to sample from.
>
> However, we do provide weak evidence against the mechanism of proof. As said in your own words:
>
> > Prompts of higher length do not increase the understanding, so misalignment rate just oscillates stochastically.
>
> As you increase the length of the prompts, there reaches a certain point where understanding isn't increased. So, were you to continue to sample sentences from the negative distribution, you wouldn't continually shift to be further negative. You'd simply oscillate. However, this isn't as rigorous of a result as it would be with a different setup like described above.
>
> Another suspicion we hadn't mentioned in the paper but would like to raise is whether $\beta$, the lower bound of KL-Divergence from additional sentences, simply may have a value of 0. If this were the case, it'd take away the power from their proofs.
>
> Take the example of a prompt of repeated A's: "AAAAA...". What the model is very likely to predict next is simply another A. However, the more A's you add to lengthen the input, it doesn't seem like it is shifting the model's probability density towards the negative sub-distribution. Also, it is unclear what negative behavior would even refer to in this case.
>
> > **R2.** Figures 2 and 6 show logit values of the "Yes" output. If I understand this correctly, these are the pre-softmax values produced by the LLM. These values are not normalized, so the plots are not very meaningful. The authors observe that "the output values generally shift to be smaller for the RLHF model compared to the base model". This is not an interesting observation, since the softmax output does not change if all inputs are shifted by a constant.
> > Regarding my original W4, I agree with Reviewer jcwj that the logit issue still has to be addressed.
>
> Thank you for the suggestion. We will revise the paper with log probability to avoid this normalization issue.
>
> Thank you for your insightful response. Exploring it has aided in concretizing our thoughts.

---

> > ### Comment · Reviewer_uiXy · 2023-11-23
> >
> > I would like to thank the authors for their response. In my view, the logit issue is still rather problematic and it has not been addressed in the revision. The evidence against Wolf et al, in authors' words, is also weak, so I will keep my score unchanged.

---

### Official Review · Reviewer_dnKS · 2023-11-09

**Soundness:** 2 fair
**Presentation:** 2 fair
**Contribution:** 2 fair
**Rating:** 3
**Confidence:** 4

**Summary:**

The paper studies the positive and negative behavior of LLMs under persona prompts. Specifically they study how prompts could be steered to produce behaviors-- good or bad and try to understand it through the lens of behavior distributions. They decompose a desired behavior into positive and negative sub-distributions and how SFT vs RLHF differ in terms of distinguishing these sub distributions.  Overall they posit that RLHF is more vulnerable than SFT when it comes to  steering towards negative behavior and thus could be more misaligned than the base model.

**Strengths:**

The paper poses two important questions which are relevant to understanding the value of alignment through RLHF. Persona prompting is a classic jailbreak approach in these LLMs and the paper tries to understand how the RLHF-ed models could be steered into positve and negative territories through persona prompting. The paper is well-written and the arguments presented are easy to follow. The paper also makes an interesting observation as to how behavior distinguishability between RLHF and SFT is different for different behaviors , thus asking us to look into the nature of behavioral annotation data collected for the RLHF step.

**Weaknesses:**

I have a couple of concerns with this paper

* The experimentation is limited to one particular model LLAMA 2. Since the authors themselves do not do any further training or finetuninng themselves, any observation that they make are subject to the design and data decisions undertaken while developing LLAMA-2. To make more concrete observations about RLHF vs SFT, I would expect to see similar behaviors across a few different class of SOTA LLMs. Otherwise, one cannot concretely say that the observations generalize across all classes of LLMs.

* Secondly, while the paper proposes some observations, I would have loved more detailed ablations into the hypotheses which cause the behavior to be observed. For instance, if one is making such an observation "RLHF training process may not have been explicitly tailored to target specific human-like psychological behaviours for avoidance", I would have loved to see an ablation where some additional human data is collected or gathered from open source data on these behaviours, further RLHF the model on this data and demonstrate that this effect disappears or is mitigated to some extent.

* Thirdly the paper presents two ideas for distinguishability -- Beta distinguishability and Wasserstein distance. They have a brief hypothesis suggesting that RLHF is not a direct likelihood maximizing loss and hence needs a different distinguishability metric. I don't quite see how introducing Wasserstein distance adds further value to the analysis  of distinguishability or the steerability results that follow later as I can see RLHF models being more steerable negatively even after a high Wasserstein distance between behaviors. Also, I do not see an explanation why other forms of distribution divergence is not good enough compared to Wasserstein.

Overall, a general theme to improve would be the limited nature of evaluations/experiments translating to relatively lesser research impact and a need for clearer narrative in terms of the questions that are asked in the paper.

**Questions:**

Thanks for your paper and your hard work in getting it to the review. I have a few questions for the authors

1. From what I understand, I think the current study is based only on the probability of the model answering Yes or No as plausible responses to the prompts. Can I understand if you introduced any extra text in the prompt to ask the model to stick to these responses only and not place its logit weight on other forms of saying something similar ? I am asking this question particularly because of your observation of using the raw logits instead of probabilities.

2. Persona prompts is one form of jailbreaks which can be used to study steerability. Did you try other forms of adversarial attacks ? I am wondering if the general notion of steerability is broader than just persona prompts.

3. In related work or other sections, can you clearly distinguish the contributions compared to Wolf et al (2023) as I see a lot of similarities particularly with respect to theory and practice around the beta-distinguishability. Particularly, it will be interesting to test-time probe the models finetuned on the behaviors in Wolf et al(2023) as another observation to see how the effects change on further finetuning for these behaviors.

4. Overall I am wondering how generalizable are these observations considering it was done on test-time probing one class of models. Would it be possible to try this analysis in few other classes of models available opensource atleast ?
There are several forms of RM training, data collection and RLHF that can be done in the wild and incase we are making generic statements about SFT vs RLHF.

5. As a follow up, can you add a limitations section to the paper explaining why or how generalizable these observations can be and what further studies need to be done to make them more generalizable

---

> ### Author Response · Authors · 2023-11-17
>
> We sincerely appreciate your feedback and take it under constructive consideration. We address each point one by one below.
>
> > **W1.** Observations are subject to the design and data decisions undertaken while developing LLAMA-2. To make more concrete observations about RLHF vs SFT, I would expect to see similar behaviors across a few different class of SOTA LLMs.
>
> The experiment is indeed limited to LLaMa 2, which was the largest open source RLHF model available. Observations we make are subject to the design and data decisions undertaken while developing LLaMa 2. We are interested in validating this on more RLHF models as they become available.
>
> > **W2.** If one is making an observation, "RLHF training process may not have been explicitly tailored to target specific human-like psychological behaviours for avoidance", I would have loved to see an ablation where some additional human data is collected to further RLHF the model on and demonstrate that this effect disappears or is mitigated to some extent.
>
> This ablation would indeed be useful. Further training the model on particular human psychological behaviors for avoidance and re-evaluating the distributions would be insightful.
>
> > **W3.** I don't quite see how introducing Wasserstein distance adds further value to the analysis of distinguishability or the steerability results. Also, I do not see an explanation why other forms of distribution divergence are not good enough compared to Wasserstein.
>
> To clarify, we couldn't use a KL-based distinguishability metric because each value was only in one subdistribution, positive *or* negative. In order to use many of the typical metrics that come to mind, you generally have to have outputs in both distributions to compare across for any given data point. As such, Wasserstein distance was the best available metric where we had a bunch of data in two distributions that weren't paired up in any particular way.
>
> We attempted to use the $\beta$-distinguishability approximation metric used in the appendix of Wolf et al. (2023) but in their work, they created two separate models to source the positive and negative distributions from that were optimized to be their respective way which lead to their $\beta$-distinguishability increasing. It was a toy scenario very tuned to their particular setup. However, we were performing test-time probing of individual production models whose behavior distributions weren't optimized in this explicit manner. As such, we felt that Wolf et al. (2023)'s approach, while important to discuss due to its inspiration of our paper, didn't best fit our needs and that rather the Wasserstein Distance which more directly approaches how far apart two distributions are would work better.
>
> *Note: Wolf et al. (2023) recently themselves in an update removed this $\beta$ approximation equation out of their work. We likely should have not used it and just worked with Wasserstein Distance.*
>
> > **Q1.** Any extra text in the prompt to ask the model to stick to these responses only and not place its logit weight on other forms of saying something similar?
>
> We did not introduce any extra text to ask the model to stick to Yes and No only in our final experiments.
>
> > **Q2.** Did you try other forms of adversarial attacks?
>
> We did not try other forms of adversarial attacks as we particularly were investigating how conditioning on additional negative sentences shifted the behavior of the model. In this framework, most adversarial attacks, which generally cleverly shift behavior via a small number of tokens, couldn't be systematically set to different lengths out to fit our needs.
>
> > **Q3.**  Can you clearly distinguish the contributions compared to Wolf et al. (2023)
>
> Wolf et al. (2023) provide the theoretical framework and do brief testing on two optimized toy models. In contrast, we do test time probing on individual production models and significantly expand on their experimentation. Wolf et al. solely calculated the toy $\beta$ approximates, whereas we attempted to see whether the consequences of their theoretical claims were actually observable in language models:
> - Do larger distinguishabilities make it easier to repeatedly recondition the model with more and more sentences towards behaving negatively?
> - Does every additional sentence sourced from the negative distribution at a minumum continually shift the model to behave more poorly?
>
> > **Q4.** Can you add a limitations section to the paper explaining why or how generalizable these observations can be and what further studies need to be done to make them more generalizable?
>
> Adding a limitations section would indeed aid in providing our thoughts on how generalizeable our results are.
>
> Thank you for these great suggestions. We will duly take them into account.

---

### Meta-Review · Area_Chair_Rpjy · 2023-12-12

**Metareview:**

### Summary

This paper investigates the susceptibility of Large Language Models (LLMs) to being steered into negative behavior using persona prompts. The paper examines the effects of Reinforcement Learning from Human Feedback (RLHF), a common alignment approach, on this susceptibility problem. The authors find that the RLHF model can be more easily steered into negative behavior than the base model, even when the negative behavior is less distinguishable from the positive behavior. These results suggest that RLHF is a less effective alignment approach for preventing LLMs from being used for malicious purposes.

### Decision

Overall, the reviewers all unanimously agreed to reject this paper. The authors have tried to address some of the concerns raised by the reviewers in the rebuttal, but many remain. For example, the claims about RLHF being more amenable to negative behaviors with persona prompts than SFT models are primarily based on some experiments with llama2. The authors also need to provide more experimental evidence on different RLHF'ed LLMs.

Based on the reviewer's comments, the paper is plagued by several weaknesses that raise doubts about its overall quality and contribution to the field. Here's a summary of the key weaknesses identified by the reviewers:

Limited scope and generality: The study focuses on a single model, LLAMA-2, and a limited set of behaviors, raising concerns about its applicability to a broader range of language models and behaviors.

Inconclusive findings and lack of explanation: The results on distinguishability are inconclusive, and the paper fails to provide a clear explanation for the contradictory conclusions using different metrics. Similarly, the steerability results are limited and lack a clear connection to the distinguishability results.

Weaker conclusion and methodology: The conclusion that RLHF is not a suitable approach for alignment is based on limited evidence and does not address potential solutions, such as refining the annotation process. The methodology in Section 4.2 uses unnormalized logits, which are not comparable across timesteps or examples.

Lack of motivation and rigor: The motivation for the "Is the following statement something you would say?" questions is not adequately supported, and the study is limited to a subset of behaviors that may not reflect the most pressing concerns regarding language model alignment. The paper relies on assertions of consensus rather than a rigorous justification for its research choices.

Similarities with existing work: The questions and analysis are very similar to previous work, raising concerns about the paper's originality and contribution.

Overall, the reviewers' comments suggest that the paper lacks sufficient rigor, clarity, and originality to warrant acceptance as a conference paper. The authors should address these weaknesses and provide more substantial evidence to support their claims and conclusions.

Here's a summary of the key weaknesses identified by the reviewers:

- **Limited scope and generality:** The study focuses on a single model, LLAMA-2, and a limited set of behaviors, raising concerns about its applicability to a broader range of language models and behaviors.

- **Inconclusive findings and lack of explanation:** The results on distinguishability are inconclusive, and the paper fails to provide a clear explanation for the contradictory conclusions using different metrics. Similarly, the steerability results are limited and lack a clear connection to the distinguishability results.

- **Weaker conclusion and methodology:** The conclusion that RLHF is not a suitable approach for alignment is based on limited evidence and does not address potential solutions, such as refining the annotation process. The methodology in Section 4.2 uses unnormalized logits, which are not comparable across timesteps or examples.

- **Lack of motivation and rigor:** The motivation for the "Is the following statement something you would say?" questions is not adequately supported, and the study is limited to a subset of behaviors that may not reflect the most pressing concerns regarding language model alignment. The paper relies on assertions of consensus rather than a rigorous justification for its research choices.

Overall, the reviewers' comments suggest that the paper lacks sufficient rigor, clarity, and originality to warrant acceptance as a conference paper. The authors should address these weaknesses and provide more substantial evidence to support their claims and conclusions.

**Justification For Why Not Higher Score:**

The reviewers unanimously agreed to reject this paper. This paper is not ready for publication yet.

**Justification For Why Not Lower Score:**

N/A

---

### Decision · Program_Chairs · 2024-01-16

Reject